# Manual Therapy Effect in Placebo-Controlled Trials: A Systematic Review and Meta-Analysis

**DOI:** 10.3390/ijerph192114021

**Published:** 2022-10-28

**Authors:** Miguel Molina-Álvarez, Alberto Arribas-Romano, Carmen Rodríguez-Rivera, Miguel M. García, Josué Fernández-Carnero, Susan Armijo-Olivo, Carlos Goicoechea Garcia

**Affiliations:** 1Escuela Internacional de Doctorado, Faculty of Health Sciences, Universidad Rey Juan Carlos, 28922 Alcorcón, Spain; 2Area of Pharmacology, Nutrition and Bromatology, Department of Basic Health Sciences, Rey Juan Carlos University, Unidad Asociada I+D+i Instituto de Química Médica (IQM) CSIC-URJC, 28922 Alcorcón, Spain; 3Department of Physical Therapy, Occupational Therapy, Rehabilitation and Physical Medicine, Rey Juan Carlos University, 28922 Alcorcón, Spain; 4High Performance Experimental Pharmacology Research Group, Rey Juan Carlos University (PHARMAKOM), 28922 Alcorcón, Spain; 5Grupo Multidisciplinar de Investigación y Tratamiento del Dolor, Grupo de Excelencia Investigadora URJC-Banco de Santander, 28922 Alcorcón, Spain; 6La Paz Hospital Institute for Health Research, IdiPAZ, 28029 Madrid, Spain; 7Faculty of Business and Social Sciences, University of Applied Sciences, 30A, 49076 Osnabruck, Germany; 8Faculties of Rehabilitation Medicine and Medicine and Dentistry, 3-48 Corbett Hall, Edmonton, AB T6G 2G4, Canada

**Keywords:** manual therapy, musculoskeletal manipulations, placebo, pain, meta-analysis

## Abstract

Purpose: Background: Evaluate whether the design of placebo control groups could produce different interpretations of the efficacy of manual therapy techniques. Methods: Nine databases were searched (EMBASE, CINAHL, PsycINFO, MEDLINE, PubMed, SCOPUS, WEB of SCIENCE, COCHRANE, and PEDro). Randomized placebo-controlled clinical trials that used manual therapy as a sham treatment on subjects suffering from pain were included. Data were summarized qualitatively, and meta-analyses were conducted with R. Results: 53 articles were included in the qualitative analysis and 48 were included in the quantitative analyses. Manipulation techniques did not show higher effectiveness when compared with all types of sham groups that were analyzed (SMD 0.28; 95%CI [−0.24; 0.80]) (SMD 0.28; 95%CI [−0.08; 0.64]) (SMD 0.42; 95%CI [0.16; 0.67]) (SMD 0.82; 95%CI [−0.57; 2.21]), raising doubts on their therapeutic effect. Factors such as expectations of treatment were not consistently evaluated, and analysis could help clarify the effect of different sham groups. As for soft tissue techniques, the results are stronger in favor of these techniques when compared to sham control groups (SMD 0.40; 95%CI [0.19, 0.61]). Regarding mobilization techniques and neural gliding techniques, not enough studies were found for conclusions to be made. Conclusions: The literature presents a lack of a unified placebo control group design for each technique and an absence of assessment of expectations. These two issues might account for the unclear results obtained in the analysis.

## 1. Introduction

Pain is not merely an accompanying symptom to a pathology, but rather a separate condition on its own [1], that has such a profound effect on the individuals that commonly ends up leading to depression [2]. Millions of Europeans [3,4] suffer from acute and chronic pain, resulting in a significant physical, emotional, and financial burden [5]. Therefore, there is an urgent need for optimal management.

Among the therapies used to manage musculoskeletal pain, manual therapy is one of the most common approaches. Manual therapy can be described as the application of a manual force applied accurately and specifically to the body, to improve pain-related symptoms and mobility in areas that are restricted or injured such as joints, connective tissues, or skeletal muscles [6]. It comprises a group of therapeutic techniques including, amongst others, soft tissue techniques (i.e., ischemic compression, pressure inhibition), neural mobilization, joint mobilization, and manipulation which can modulate pain, especially in a short time [7,8,9]. On the other hand, neural mobilization or neurodynamic techniques are used for analgesic purposes and are intended to improve adaptability, reduce mechanosensitivity and activate analgesic mechanisms through mechanical stimulation of the nerves through elongation and sliding [10,11]. Alternatively, pursuing the same aim, passive specific joint mobilizations can be carried out at low velocity and high amplitude of movement or high velocity and low amplitude of movement (manipulation) of articulation [12]. Passive-specific mobilizations at low velocity and high amplitude of movement consist of rhythmic, passive, and smooth movements on a joint, where strength and amplitude are controlled according to the tolerance level of the patient [13]. The manipulation technique, on the other hand, consists of making a fast and controlled movement at the end of the range of a joint, producing cavitation [14].

Although manual therapies are commonly effective in the management of musculoskeletal conditions; several authors [15,16,17] believe that their positive effects result from the placebo effect, and not necessarily from their supposed therapeutical effect. The placebo effect is defined as a genuine positive psychological and/or physiological effect, occurring in humans and other animals, attributable single-handedly to the knowledge of receiving a substance or undergoing a procedure, but not due to the inherent power of that substance or procedure [18]. Placebo effects can be generated by previous experience, perception, information acquired about the treatment, and by active integration of all of these with the sensory information patients learn during a treatment [19,20]. Thus, in the handling of pain, the placebo effect is especially important. Nevertheless, it is a combination of such physical discomfort associated with an emotional response to it, that depends on previous experience, cultural background, emotions, and other neurobiological variations, exactly in the same way as the placebo effect does [21].

When using sham treatments as control groups to determine the effectiveness of a certain active therapy, such as manual therapies, it is important to design them well. They must be non-active and harmless actions that by themselves should not produce any physical improvement/deterioration of the patient’s condition, residing here the importance of including a proper sham for each and every one of the evaluated manual therapies. Moreover, the blinding of the patients must be ensured to render a proper and adequate placebo intervention and reliable results [22,23].

Placebo-controlled Randomized Clinical Trials (RCTs) are studies that use a sham control group to simulate the treatment to be evaluated. As already stated, the purpose of this group is to provide a reliable comparator, so that the effect of a substance (i.e., placebo) or procedure (i.e., sham) can be exclusively attributed to itself [24,25]. In addition to high-quality sham control, other methodological factors such as the blinding of both placebo and treatment groups (regardless of the subjects´ beliefs [26]), randomization allocation concealment, and adequate control of biases are of importance when determining the accuracy of treatment effects. From a methodological point of view, when an RCT does not guarantee that the sham control group considers all non-specific effects of the treatment, fails to ensure that the placebo is inert, or fails to confirm that its control subjects were efficiently blinded, it makes it difficult to make confident interpretations of the results of the trial [26]. Therefore, it is important to determine whether the sham groups used in the literature have been designed to provide reliable comparisons. However, we could not identify any previous review regarding this subject.

Thus, the objective of the study was to evaluate whether the design of placebo control groups could produce different interpretations of the efficacy of manual therapy techniques.

## 2. Materials and Methods

The systematic review and meta-analysis were carried out following the Cochrane Handbook for Systematic Review of Intervention [27] and the guide “Preferred Reporting Items for Systematic Reviews, PRISMA” [28,29]. The protocol was registered at PROSPERO with ID: CRD42020157468.

### 2.1. Literature Search

The following databases were searched until 11 December 2019, from inception: EMBASE, CINAHL, PsycINFO, MEDLINE, PubMed, SCOPUS, WEB of SCIENCE, COCHRANE, and PEDro. The search strategy is based on the combination of medical terms (Mesh) and keywords, regarding the following scheme: Population: >18 years old subjects; Intervention: manual therapy; Comparative group: placebo OR sham; Outcome: pain scale. Subsequently, the search results were managed in Mendeley, proceeding with duplicate removal.

### 2.2. Study Selection

Types of Studies. Parallel RTCs carried out in humans were included, however, cross-over designs were excluded to avoid the carry-over effect. In addition, letters, reports, and abstracts from congresses were also excluded.Types of population. All studies which were conducted with subjects suffering from pain were included, regardless of the characteristics of the participants. Nevertheless, when the pain was artificially induced in the subjects (i.e., experimental pain models), studies were excluded.Types of interventions. It was compulsory for the studies to be included, to have used as a unique technique one of the following manual therapy techniques: manipulation, mobilization, soft tissue techniques, and neural mobilization in at least one intervention group. Intervention groups that were complemented with other techniques (i.e., electrotherapy, exercise) were not included, to allow the analysis of the individual effect of each technique. In the same way, therapies that involved active subject participation (i.e., exercise) were excluded. Finally, alternative medicine therapies such as reflexology or holistic treatments were excluded.Types of comparison. To be included in the systematic review, the studies had to include at least one sham group, termed by the authors as: a placebo group, or sham group. If the subjects did not receive any intervention in the placebo or sham group, it was not considered a valid sham comparison group.Types of outcomes. Pain intensity had to be assessed with rating scales such as the Visual Analog Scale (VAS), Numerical Rating Scale (NRS), Numerical Pain Rating Scale (NPRS), or similar.Time points. For the quantitative analysis, the pre-treatment assessment and the first available post-treatment assessment were chosen.

### 2.3. Selection of Studies

Firstly, duplicates were removed and transferred to a Microsoft Excel spreadsheet; afterward, articles were screened through title and abstract for eligibility, and finally, a full-text evaluation of the studies was performed. Every article identified in the different databases was reviewed independently by two authors (C.R.-R. and A.A.-R.). The authors had to reach a consensus on whether a trial should be included. If the authors did not reach a consensus, a third author (M.M.-A.), who participated in the process, made the final decision.

### 2.4. Data Extraction

Required data was extracted by two independent reviewers (C.R.-R. and A.A.-R.) using an excel sheet. From the included studies, data such as author(s), year of publication, participant characteristics (sample size, gender, age, and pathology), group description, outcome measurements, and results, were collected. After data compilation, both authors had to reach a consensus on each item of the extracted data, and in case of disagreement between the authors, a third author (M.M.-A.) made the final decision.

### 2.5. Blinding Quality Assessment

Since the aim of the study was to evaluate whether the design of placebo control groups could produce different interpretations of the efficacy of manual therapy techniques, the blinding of the subjects was assessed. Assessment of blinding quality was performed through the Risk of Bias (RoB) of the Cochrane Handbook for Systematic Reviews of Interventions to assess bias [30], independently by two reviewers (C.R.-R. and A.A.-R.). Any discrepancies in quality ratings were solved by discussion. If consensus could not be reached, a third member of the review team (M.M.-A.) acted as an arbitrator.

The analysis of blinding was complemented, as other authors have described [31], with the evaluation of other domains of the RoB tool such as random sequence generation, allocation concealment, participant blinding, and detection bias. The overall RoB for the above key domains was named as: “Adequate” if a trial was rated low risk (all key domains as low risk or at most with one unclear risk domain); or “Not adequate” if a study was considered an unclear risk (unclear risk in more than one domain) or high risk (high risk in at least one domain). Furthermore, research personnel blinding and therapist blinding proved to be impossible to achieve due to the techniques´ procedure, therefore these domains were not taken into account.

All the studies which did not perform a simulation of the intervention groups (i.e., fake massage as a control of manipulation interventions) were cataloged as high risk in “participant blinding”. Hence, studies that applied detuned devices or simulations of other techniques were assumed to not be valid. We considered that these placebo control groups could interfere with the expectations of the participants, patients undergoing a manual therapy technique do not have the same experiences as patients undergoing an electrotherapy-based treatment. Placebo is defined as a genuine positive psychological and/or physiological effect, occurring when undergoing a procedure, but it is not due to the inherent powers of that procedure [18], thus, the placebo groups have to guarantee: the replication of the protocol, with the objective that the subjects have the same expectations and undergo the same subjective experience; harmlessness, the sham group should not produce a greater effect than the technique, at most, the same effect.

To assess inter-rater reliability, the kappa coefficient (κ) was calculated to assess reliability prior to any consensus and the inter-rater reliability was estimated by using the kappa coefficient (κ) considering that κ > 0.7 indicates a high level of agreement among reviewers; κ of 0.5–0.7 indicates a moderate level of agreement; and a κ < 0.5 indicates a low level of agreement [32]. Disagreements on the quality assessment of RoB were resolved by consensus with a third independent reviewer.

### 2.6. Data Analysis and Synthesis

Data was analyzed qualitatively and compiled in evidence tables. The quantitative analyses were conducted with the metafor package in R [33,34]. The results were analyzed using the Standardized Mean Difference (SMD) with its 95% Confidence Interval (CI). Since different scales were used to measure pain, the SMD Hedge’s g was used. All forest plots reported results on pain intensity. The effect size was evaluated following Cohen’s magnitude criteria for rehabilitation treatment effects: d = 0.14–0.31 “small” effect size; d = 0.31–0.61 “medium” effect size; and d > 0.61 “large” effect size [35]. Effects were interpreted as statistically significant when *p* < 0.05. The inverse variance statistical analysis method was used. Combined results analysis was carried out using the random effect model, as opposed to the fixed effect model, incorporating the variance in each study and between studies [27].

### 2.7. Assessment of Heterogeneity

Heterogeneity among trials was assessed using heterogeneity statistics (e.g., Cochrane’s Q test and I^2^ statistic). It was considered that Q *p*-values < 0.10 indicate a significant heterogeneity [35]. In addition, I^2^ describes the percentage of variability due to heterogeneity, and not to random error or chance. I^2^ values between 0% and 30% were considered to have null to low heterogeneity; between 30% and 50%, medium heterogeneity; between 50% and 75%, moderate heterogeneity; and over 75%, high heterogeneity [36].

### 2.8. Subgroup Analysis

After the overall analysis, subgroup analyses were performed according to blinding adequacy and type of control sham group. When looking at the adequacy of blinding, two groups were compared: studies with adequate blinding versus those with inadequate blinding. On the other hand, when studying the effect of types of sham groups, studies were grouped into the following categories: detuned devices, “therapeutic” (i.e., soft massage), and different types of simulation of the techniques (i.e., simulation with movement).

### 2.9. Assessment of the Level of Evidence

To assess the overall quality of evidence in the outcomes of the meta-analysis, the Grading of Recommendations Assessment, Development and Evaluation (GRADE) [37] approach was implemented. The quality of evidence was categorized as follows: high: the true effect lies close to the estimate of the effect; moderate: the true effect is likely to be close to the estimate of the effect, but there is a possibility that it is substantially different; low: the true effect may be substantially different from the estimate of the effect; very low: the true effect is likely to be substantially different from the estimate of the effect.

## 3. Results

### 3.1. Selection of Trials

The literature search produced 8753 results, of which 5163 were duplicated. After the first screening, a total of 3091 studies were excluded. Following the whole text reading, 445 articles were excluded and 53 remained. Finally, these 53 articles were included in the qualitative analysis; of which only 48 were included in the quantitative analysis, as they did present the necessary measures to be analyzed (Figure 1). Inter-rater reliability for assessing the risk of bias was very high (κ = 0.94). The principal characteristics of the trials are available in Table 1.

### 3.2. Trials Using Manipulation Techniques

#### 3.2.1. Participants

A total of 1827 subjects were included in the qualitative analysis of the manipulation techniques, 1226 of which were females. The average age of participants was 34.05 years old. Eight trials (27%) included acute or chronic neck pain [38,39,40,41,42,43,44,45]; seven (23%) acute or chronic low back pain [46,47,48,49,50,51,52]; six trials (20%) shoulder pain [53,54,55,56,57,58]; three (10%) headache [59,60,61]; two (7%) primary dysmenorrhea [62,63]; and trials looked at thoracic spine pain (3%) [64]; patellofemoral pain syndrome (3%) [65]; temporomandibular disorder (3%) [66]; and cervical radiculopathy (3%) [67].

#### 3.2.2. Intervention Groups

The trials had to perform at least one manipulation in one intervention group and this procedure had to be the only one the subjects received. Most of the trials performed one session of treatment (63%) [38,39,40,41,42,43,44,45,50,52,54,55,57,58,59,63,65,66,67], the trial conducted by Senna et al. [51] the longest one, conducted over a 10-month period and completing 48 sessions of manipulations during this time. Concerning the pain emplacement and manipulation location: five trials (63%) which presented subjects suffering from neck pain performed cervical spine manipulation [38,39,40,43,44] and three (37%) applied thoracic spine manipulation [41,42,45]; in low back pain trials applied lumbar manipulation in all interventions [46,47,48,49,50,51,52]; for subjects who suffered from shoulder pain, the authors designed interventions with thoracic spine manipulation [54,55,56,57,58] (83%) and shoulder manipulation in one trial (17%) [53]; for headache, the subjects received cervical spine manipulation [59,60,61]; for thoracic spine pain, participants underwent thoracic spine manipulation [64]; for patellofemoral pain syndrome the researchers applied lumbopelvic manipulation [65]; for temporomandibular disorder, one trial tried thoracic spine manipulation [66]; and in patients who suffered cervical radiculopathy, cervical spine manipulation was performed [67].

#### 3.2.3. Placebo

The sham groups described by the authors showed two principal trends: some authors designed interventions applying detuned devices (10%) [40,53,64], and several trials (80%) performed a simulation of the techniques applied in the intervention group [38,41,42,43,44,45,46,47,48,49,51,52,54,55,56,57,58,59,60,62,63,65,66,67]. Within the second group, the authors planned different strategies: the most common (62.5%) was the strategy described by Cleland et al. [68] and Michener et al. [69] simulating the procedure but without the rapid application of motion characteristic of the manipulation technique [41,42,45,48,49,52,54,55,56,57,58,63,65,66,67]; the second most common approach (37%), was to simulate the technique and even reproduce the rapid application of motion but without inducing the thrust [38,43,44,46,47,51,59,60,62], as proposed, among others, Vernon et al. [70]. The exception from these studies was the trial conducted by Espí-López et al. [61], which did not simulate the technique nor used a detuned device, despite the control group undergoing the artery test. In addition, Kawchuk et al. [50] did not perform any intervention since the subjects were under the effects of general anesthesia, in this work, the placebo was performed through verbal suggestion conditioning by telling patients they had received manipulation when in fact they had received nothing. Finally, Haas et al. [39] performed a cervical spine manipulation in a “sham endfeel finding”.

**Table 1 ijerph-19-14021-t001:** Studies characteristics.

**Manipulation Techniques**							
**Author/Year**	**Participants**	**Pathology**	**Groups**	**Intervention Group Description**	**Placebo Group Description**	**Pain** **Outcome**	**Measurements**
García-Perez-Juana et al. [38]	n = 54 Female (n = 42) Age (years): 37 (8)	Neck pain (Chronic)	Manipulation I (n = 18) Manipulation II (n = 18) Placebo group (n = 18) 1 session	Manipulation I: One midcervical spine manipulation on the right side was performed. Manipulation II: One midcervical spine manipulation on the left side was performed.	Simulation of the technique and even reproduced the rapid application of motion but without inducing the thrust.	NPRS	Preintervention 1-week post-intervention
Haas et al. [39]	n = 99 Female (n = 63) Age (years): 42.55 (SD Not available)	Neck pain	Manipulation (n = 47) Placebo group (n = 52) 1 session	Cervical manipulation was performed in the “endfeel finding”	Real cervical manipulation was performed in a “sham endfeel finding”.	VAS	Preintervention Post-intervention
Pikula [40]	n = 36 Female (n = 28) Age (years): 42.10 (SD Not available)	Unilateral neck pain	Manipulation I (n = 12) Manipulation II (n = 12) Placebo group (n = 12) 1 session	Manipulation I: cervical manipulation was applied to the same side as the pain Manipulation II: cervical manipulation was applied to the side opposite the pain	A detuned ultrasound was used.	VAS	Preintervention Post-intervention
Pires et al. [41]	n = 32 Female (n = 32) Age (years): 24.7 (SD Not available)	Neck pain (Chronic)	Manipulation (n = 16) Placebo group (n = 16) 1 session	Upper thoracic spine manipulation was performed on vertebra T1.	Simulation of the procedure but without the rapid application of motion	VAS	Preintervention Post-intervention 48–72 h post-intervention
Sillevis et al. [42]	n = 100 Female (n = 77) Age (years): 44.77 (SD Not available)	Neck pain	Manipulation (n = 50) Placebo group (n = 50) 1 session	Thoracic manipulation in vertebral level T3-T4 was performed in a supine position.	Simulation of the procedure but without the rapid application of motion	VAS	Preintervention Post-intervention
Valera-Calero et al. [43]	n = 83 Female (n = 51) Age (years): 36.61 (SD Not available)	Neck pain (Chronic)	Manipulation (n = 28) Mobilization (n = 28) Placebo group (n = 27) 1 session	Cervical manipulation was performed on vertebral levels C5–C6.	Simulation of the procedure but without the rapid application of motion	VAS	Preintervention Post-intervention 1-week post-intervention
Vernon et al. [44]	n = 64 Female (n = 34) Age (years): 38.55 (SD Not available)	Neck pain	Manipulation (n = 32) Placebo group (n = 32) 1 session	Cervical manipulation was performed	Simulation of the technique and even reproduced the rapid application of motion but without inducing the thrust.	NPRS	Preintervention 5 min post-intervention 15 min pos-intervention
Weber et al. [45]	n = 24 Female (n = 16) Age (years): 38.0 (15.1)	Neck pain (Acute or subacute)	Manipulation (n = 12) Placebo group (n = 12) 1 session	Thoracic spinal manipulation was performed in the mid-thoracic spine.	Simulation of the procedure but without the rapid application of motion	NPRS	Preintervention Post-intervention
Bialosky et al. [46]	n = 110 Female (n = 77) Age (years): 31.68 (11.85)	Low back pain	Manipulation (n = 28) Placebo group I (n = 27) Placebo group II (n = 27) Control group: no treatment (n = 28) 5 sessions for 2 weeks	Lumbar manipulation was performed 2 times on each side.	Placebo group I: simulation of the technique and even reproduced the rapid application of motion but without inducing the thrust. Placebo group II: same simulation + increased expectations.	NRS	Preintervention Post-intervention
Bond et al. [47]	n = 29 Female: (n = 11) Age (years): 23.86 (5.74)	Low back pain (Chronic)	Manipulation (n = 14) Placebo group (n = 15) 3 sessions per week for 2 weeks	Two lumbopelvic manipulations were performed on both sides of the pelvis, alternating between the left and right sides.	Simulation of the technique and even reproduced the rapid application of motion but without inducing the thrust.	NPRS	Preintervention < 1-week post-intervention
Didehdar et al. [48]	n= 25 Female (n = 11) Age (years): 36.6 (SD Not available)	Low back pain (Chronic)	Manipulation (n = 10) Placebo group (n = 15) 3 sessions in 6 days	Sacroiliac and lumbar spine manipulations were performed.	Simulation of the procedure but without the rapid application of motion	NRS	Preintervention 5 weeks post-intervention
Elleuch and Ghroubi [49]	n = 85 Female (n = 68) Age (years): 38.34 (10.2)	Low back pain (Chronic)	Manipulation (n = 50) Placebo group (n = 35) 1 session per week for 4 weeks	4 manipulations to the low back were performed	Simulation of the procedure but without the rapid application of motion	VAS	Preintervention 1-month post-intervention 2 months post-intervention
Kawchuk et al. [50]	n = 6 Female (n = 2) Age (years): 36.5 (SD Not available)	Low back pain (Acute)	Manipulation (n = 3) Placebo group (n = 3) 1 session	One lumbar manipulation was performed while the patients were under the effects of general anesthesia	In the placebo group, the patients were told that they were going to receive a manipulation, but no treatment was performed while the patients were under the effects of general anesthesia.	NRS	Preintervention 30 min post-recovery
Senna and MacHaly [51]	n = 88 Female (n = 22) Age (years): 41.41 (SD Not available)	Low back pain (Chronic)	Manipulation I (n = 27) Manipulation II (n = 26) Placebo group (n = 40) Manipulation I and sham manipulation: 3 sessions per week for 1 month. Manipulation II: same sessions plus after the first month 1 session every 2 weeks for the next 9 months.	Lumbar manipulation was performed.	Simulation of the technique and even reproduced the rapid application of motion but without inducing the thrust.	VAS	Preintervention Post-intervention 3 months post-intervention 6 months post-intervention 10 months post-intervention
Vieira-Pellenz et al. [52]	n = 40 No females Age (years): 38 (9.14)	Degenerative lumbar disease	Manipulation (n = 20) Placebo group (n = 20) 1 session	Lumbar manipulation was performed.	Simulation of the procedure but without the rapid application of motion	VAS	Preintervention Post-intervention
Atkinson et al. [53]	n = 60 Female (n = 17) Age (years): 41.76 (SD Not available)	Shoulder pain (Rotator cuff tendinopathy)	Manipulation (n = 30) Placebo group (n = 30) 6 sessions for 2 weeks	Shoulder manipulation was performed.	A detuned laser was used.	NRS	Preintervention Post-intervention third session Post-intervention sixth session
Conte-da-Silva et al. [54]	n = 60 Female (n = 41) Age (years): 45.26 (SD Not available)	Shoulder pain	Manipulation (n = 30) Placebo group (n = 30) 1 session	Thoracic manipulation was performed in prone.	Simulation of the procedure but without the rapid application of motion	VAS	Preintervention Post-intervention
Haik et al. [55]	n = 50 Female (n = 18) Age (years): 31.8 (10.9)	Shoulder impingement syndrome	Manipulation (n = 25) Placebo group (n = 25) 1 session	Thoracic manipulation was performed in a seated position.	Simulation of the procedure but without the rapid application of motion	NPRS	Preintervention Post-intervention
Haik et al. [56]	n = 61 Female (n = 23) Age (years): 31.9 (SD Not available)	Shoulder impingement syndrome	Manipulation (n = 27) Placebo group (n = 28) 2 sessions for 1 week	Thoracic manipulation was performed in a seated position.	Simulation of the procedure but without the rapid application of motion	NPRS	Preintervention first session Preintervention second session Post-intervention second session 3–4 days post-intervention
Kardouni et al. [57]	n = 52 Female (n = 24) Age (years): 32.0 (SD Not available)	Shoulder impingement syndrome	Manipulation (n = 26) Placebo group (n = 26) 1 session	6 thoracic manipulations were performed per participant: 2 to the lower, 2 to the middle, and 2 to the upper thoracic spine.	Simulation of the procedure but without the rapid application of motion	NPRS	Preintervention Post-intervention 24–48 h post-intervention
Kardouni et al. [58]	n = 45 Female (n = 23) Age (years): 31.15 (SD Not available)	Shoulder impingement syndrome	Manipulation (n = 24) Placebo group (n = 21) 1 session	6 thoracic manipulations were performed per participant: 2 to the lower, 2 to the middle, and 2 to the upper thoracic spine.	Simulation of the procedure but without the rapid application of motion	NPRS	Preintervention Post-intervention 24–48 h post-intervention
Borusiak et al. [59]	n = 52 Female (n = 31) Age (years): 11.6 (2.3)	Cervicogenic headache	Manipulation (n = 28) Placebo group (n = 28) 1 session	Cervical manipulation was performed	Simulation of the technique and even reproduced the rapid application of motion but without inducing the thrust.	VAS	Preintervention 2 months post-intervention
Chaibi et al. [60]	n = 98 Female (n = 83) Age (years): 39.8 (SD Not available)	Migraine	Manipulation (n = 34) Placebo group (n = 34) Control group (n = 29) 12 sessions for 3 months	Manipulation was performed on an unspecific vertebral level, wherever the therapist considered it necessary.	Simulation of the procedure but without the rapid application of motion	NRS	Preintervention Post-intervention 3 months post-intervention 6 months post-intervention 12 months post-intervention
Espí-López et al. [61]	n = 84 Female (n = 68) Age (years): 39.76 (SD Not available)	Tension-type headache	Soft tissue technique (n = 20) Manipulation (n = 22) *Combined treatment intervention (n = 20)* Placebo group (n = 22) 1 session per week for 4 weeks	Occiput-atlas-axis bilateral manipulation was performed in the manipulation group.	The control group underwent the artery test and remained for 10 min in a resting position.	VAS	Preintervention Post-intervention 1-month post-intervention
Hondras et al. [62]	n = 138 Female (n = 138) Age (years): 30.4 (SD Not available)	Primary dysmenorrhea	Manipulation (n = 69) Placebo group (n = 69) 1 session on day 1 of the menstrual cycle for 4 months	Manipulation was performed in all clinically relevant vertebral levels between T10-L5 and both sacroiliac joints.	Simulation of the technique and even reproduced the rapid application of motion but without inducing the thrust.	VAS	Preintervention first session Post-intervention first session Preintervention second session Post-intervention second session
Molins-Cubero et al. [63]	n = 40 Female (n = 40) Age (years): 30 (6.10)	Primary dysmenorrhea	Manipulation (n = 20) Placebo group (n = 20) 1 session	Manipulation of bilateral sacroiliac joint and vertebral level L5-S1 was performed.	Simulation of the procedure but without the rapid application of motion	VAS	Preintervention Post-intervention
Lehtola et al. [64]	n = 109 Female (n = 109) Age (years): 42.5 (SD Not available)	Thoracic spine pain	Manipulation (n = 37) *Acupuncture (n = 35)* Placebo group (n = 37) 4 sessions in 3 weeks	Thoracic manipulation was performed in a supine position, between the vertebral levels T3-T8 many times as the therapist considered.	Detuned electrotherapy with suction cups.	VAS	Preintervention first session Preintervention second session 1 week post-intervention
Motealleh et al. [65]	n = 28 Female (n = 16) Age (years): 26.5 (SD Not available)	Patellofemoral pain syndrome	Manipulation (n = 14) Placebo group (n = 14) 1 session	Lumbopelvic manipulation was performed once with the patient positioned supine.	Simulation of the procedure but without the rapid application of motion	VAS	Preintervention Post-intervention
Packer et al. [66]	n = 32 Female (n = 32) Age (years): 24.7 (SD Not available)	Temporomandibular disorder	Manipulation (n = 16) Placebo group (n = 16) 1 session	Upper thoracic spine manipulation was performed on vertebra T1.	Simulation of the procedure but without the rapid application of motion	VAS	Preintervention Post-intervention 48–72 h post-intervention
Young et al. [67]	n = 43 Female (n = 29) Age (years): 46 (SD Not available)	Cervical radiculopathy	Manipulation (n = 22) Placebo group (n = 21) 1 session	Thoracic spine manipulation was performed in the vertebral levels C7-L3 and T4-T9.	Simulation of the procedure but without the rapid application of motion	NPRS	Preintervention Post-intervention 48–72 h post-intervention
**Soft Tissue Techniques**							
**Author/Year**	**Participants**	**Pathology**	**Groups**	**Intervention Group Description**	**Placebo Group Description**	**Pain Outcome**	**Measurements**
Espí-López et al. [61]	n = 84 Female (n = 68) Age (years): 39.76 (SD Not available)	Tension-type headache	Soft tissue technique (n = 20) Manipulation (n = 22) *Combined treatment intervention (n = 20)* Placebo group (n = 22) 1 session per week for 4 weeks	Soft tissue technique consisted of suboccipital inhibition for 10 min.	The subjects underwent the artery test and remained for 10 min in a resting position.	VAS	Preintervention Post-intervention 1-month post-intervention
Antolinos-Campillo et al. [71]	n = 40 Female (n = 17) Age (years): 34 (3.6)	Cervical whiplash	Soft tissue technique (n = 20) Placebo group (n = 20) 1 session	Suboccipital muscle inhibition was performed for 4 min.	Active movement of flexion/extension of the hip and knee joints for 4 min.	VAS	Preintervention Post-intervention
Blikstad and Gemmell [72]	n = 45 Female (n = 25) Age (years): 23.8 (1.153)	Non-specific neck pain	*Activator device (n = 15)* Soft tissue technique (n = 15) Placebo group (n = 15) 1 session	Myofascial band therapy consisted of a slow stroking motion from the lateral shoulder to the mastoid process along the upper trapezius muscle.	A detuned ultrasound was used.	NRS	Preintervention Post-intervention
Briem et al. [73]	n = 40 Female (n = 31) Age (years): 34.7 (SD Not available)	Neck pain	Soft tissue technique (n = 20) Placebo group (n = 20) 1 session	Inhibitive distraction was applied for 3 to 3.5 min onto the suboccipital musculotendinous structures.	Gently positioning the hands on the massage area in the absence of movement or pressure.	NPRS	Preintervention Post-intervention
Buttagat et al. [74]	n = 50 Female (n = 43) Age (years): 22.24 (SD Not available)	Neck pain (Myofascial trigger points)	Soft tissue technique (n = 25) Placebo group (n = 25) 1 session	Massage was performed for 30 min on the upper neck and upper back.	A detuned microwave diathermy was used for 30 min.	VAS	Preintervention Post-intervention
Capó-Juan et al. [75]	n = 75 Female (n = 60) Age (years): 38.28(0.68)	Neck pain (caused by sternocleidomastoid)	*Kinesiotape (n = 25)* Soft tissue technique (n = 25) Placebo group (n = 25) 1 session	Pressure release was applied on the active myofascial trigger points in the stermocleidomastoid.	Application of algometric bilateral pressure for the placebo group.	NRS	Preintervention 1-week post-intervention
Gemmell et al. [76]	n = 45 Female (not available) Age (years): 23.67 (SD Not available)	Neck pain (caused by upper trapezius)	Soft tissue technique I (n = 15) Soft tissue technique II (n = 15) Placebo group (n = 15) 1 session	Soft tissue technique I: Ischemic compression was applied with the thumb to the upper trapezius myofascial trigger point for 30 s to 1 min. Soft tissue technique II: Pressure release was applied with the thumb to the upper trapezius myofascial trigger point for 90 s.	A detuned ultrasound was used.	VAS	Preintervention Post-intervention
Chatchawan et al. [77]	n = 72 Female (n = 55) Age (years): 27.37 (SD Not available)	Chronic tension-type and migraine headaches.	Soft tissue technique (n = 36) Placebo group (n = 36) 9 sessions for 3 weeks	The massage group consisted of 25 min of massage on the upper neck, and upper back.	A detuned ultrasound was used.	VAS	Preintervention Post-intervention 3 weeks post-intervention 9 weeks post-intervention
Ferragut-Garcías et al. [78]	n = 97 Female (n = 78) Age (years): 39.4 (SD Not available)	Tension-type headache	Soft tissue technique (n = 23) Neural technique (n = 25) *Combined treatment intervention (n = 25)* Placebo group (n = 24) 2 sessions in the first 2 weeks and 1 session per week in the next 2 weeks	Soft tissue techniques were performed on the sternocleidomastoid muscle, temporal muscle, suboccipital musculature, masster muscle, and upper trapezius muscle. The protocol lasted 15 min.	Superficial massage was performed for 15 min using ultrasound gel to minimize skin stimulation.	VAS	Preintervention Post-intervention 15 days post-intervention 30 days post-intervention
Arguisuelas et al. [79]	n = 54 Female (n = 33) Age (years): 46.6 (10.3)	Low back pain (Chronic)	Soft tissue technique (n = 26) Placebo group (n = 26) 2 sessions per week for 2 weeks	Massotherapy was applied to lumbar paravertebral muscles, thoracolumbar fascia, quadratus lumborum, and psoas muscle. (45 min)	Gently positioning the hands on the massage area in absence of movement or pressure.	VAS	Preintervention Post-intervention 10 weeks post-intervention
Celenay et al. [80]	n = 63 Female (n = 51) Age (years): 52 (SD Not available)	Low back pain (Chronic)	Soft tissue technique (n = 21) Placebo group (n = 21) *Control group: Standardized physiotherapy program (n = 21)* 5 sessions per week for 3 weeks	Connective tissue massage was performed for 10 to 15 min each session on the lumbar area.	In the sham connective tissue massage general slow and slight strokes and effleurage on the lower back area were applied using no specific technique and specific muscles.	VAS	Preintervention Post-intervention
Field et al. [81]	n = 20 Female (n = 16) Age (years): 47 (SD Not available)	Chronic fatigue syndrome	Soft tissue technique (n = 10) Placebo group (n = 10) 2 sessions per week for 5 weeks	Full body massage was performed for 30 min.	A detuned TENS was used.	VAS	Preintervention first session Post-intervention first session Preintervention last session Post-intervention last session
Nourbakhsh and Fearon [82]	n = 23 Female (n = 9) Age (years): 52.55 (SD Not available)	Lateral epicondylitis	Soft tissue technique (n = 11) Placebo group (n = 12) 6 sessions in 2–3 week	Inhibition pressure release for 30 s to 2 min was performed in the most painful myofascial trigger point	Gently positioning the hands on the massage area in absence of movement or pressure.	NRS	Preintervention Post-intervention
Tanaka et al. [83]	n = 32 Female (n = 29) Age (years): 81 (SD Not available)	Knee Osteoarthritis	Soft tissue technique (n = 16) Placebo group (n = 16) 1 session	Continuous compression for 5 min was performed on the anterior and distal portions of the medial thigh.	Gently positioning the hands on the massage area in absence of movement or pressure.	VAS	Preintervention Post-intervention
**Mobilization Techniques**							
**Author/Year**	**Participants**	**Pathology**	**Groups**	**Intervention Group Description**	**Placebo Group Description**	**Pain Outcome**	**Measurements**
Valera-Calero et al. [43]	n = 83 Female (n = 51) Age (years): 36.61 (SD Not available)	Neck pain (Chronic)	Manipulation (n = 28) Mobilization (n = 28) Placebo group (n = 27) 1 session	Cervical mobilization was performed on vertebral level C5-C6 for three sets of one min.	Simulation of the manipulation but without the rapid application of motion	VAS	Preintervention Post-intervention 1-week post-intervention
Snodgrass et al. [84]	n = 64 Female (n = 48) Age (years): 33.4 (SD Not available)	Neck pain (Chronic)	Mobilization I (n = 22) Mobilization II (n = 21) Placebo group (n = 21) 1 session	Mobilization I: low force posterior-to-anterior cervical mobilization (31± 1 N) was performed in 3 sets of 1 min on the most painful vertebral level. Mobilization II: high force posterior-to-anterior cervical mobilization (89 ± 3 N) was performed in 3 sets of 1 min on the most painful vertebral level.	A detuned laser was used.	VAS	Preintervention Post-intervention 4 days post-intervention
Kogure et al. [85]	n = 179 Female (n = 111) Age (years): 59.8 (13.1)	Low back pain (Chronic)	Mobilization (n = 90) Placebo group (n = 89) 1 session per month for 6 months	Sacroiliac mobilizations consisting of upward gliding, downward gliding, superior distraction, and inferior distraction were performed for 15–20 min.	The therapist simulated the treatment, giving a light force on the joint, but did not actually produce movement.	VAS	Preintervention first session Preintervention second session Preintervention third session Preintervention fourth session Preintervention fifth session Preintervention sixth session
Krekoukias et al. [86]	n = 75 Female (n = 33) Age (years): 47.51 (SD Not available)	Low back pain	Mobilization (n = 25) Placebo group (n = 25) *Conventional physiotherapy (Stretching exercises, TENS, and massage) (n = 25)* 1 session per week for 5 weeks	Spinal lumbar mobilization for 10 min: passive accessory intervertebral movements and passive physiological intervertebral movements.	The therapist mimicked the grip and the procedure of mobilization without performing any force.	NPRS	Preintervention Post-intervention
Silva et al. [87]	n = 38 Female (n = 22) Age (years): 40.8 (2.0)	Ankle injury (subacute or chronic)	Mobilization (n = 19) Placebo group (n = 19) 3 sessions per week for 2 weeks	Cyclic and rhythmic mobilizations were applied to the talus in the anteroposterior direction.	The therapist simulated the treatment, giving a light force on the joint, but did not actually produce movement.	VAS	Preintervention Post-intervention first session Post-intervention sixth session 2 weeks post-intervention
La Touche et al. [88]	n = 32 Female (n = 21) Age (years): 33.87 (SD Not available)	Cervico-craniofacial pain	Mobilization (n = 16) Placebo group (n = 16) 3 sessions in 2 weeks	Mobilization for 6 min on the 3 upper cervical segments.	The therapist mimicked the grip and the procedure of mobilization without performing any force.	VAS	Preintervention first session Post-intervention first session Preintervention second session Post-intervention second session Preintervention third session Post-intervention third session
Pecos-Martin et al. [89]	n = 34 Female (n = 19) Age (years): 24 (3)	Thoracic spine pain	Mobilization (n = 17) Placebo group (n = 17) 1 session	Mobilization on T7 was performed using a pisiform grip three times for 1 min, with a 20-s rest between sets.	The therapist simulated the treatment, giving a light force on the joint, but did not actually produce movement.	NRS	Preintervention Post-intervention
Surenkok et al. [90]	n = 39 Female (n = 22) Age (years): 54.30 (14.16)	Shoulder pain	Mobilization (n = 13) Placebo group (n = 13) Control group: no treatment (n = 13) 1 session	Sets of 10 repetitions of scapular mobilization were performed with a rest interval of 30 s between sets.	The therapist simulated the treatment, giving a light force on the joint, but did not actually produce movement.	VAS	Preintervention Post-intervention
**Neural Techniques**							
**Author/Year**	**Participants**	**Pathology**	**Groups**	**Intervention Group Description**	**Placebo Group Description**	**Pain Outcome**	**Measurements**
Ferragut-Garcías et al. [78]	n = 97 Female (n = 78) Age (years): 39.4 (SD Not available)	Tension-type headache	Soft tissue technique (n = 23) Neural technique (n = 25) *Combined treatment intervention (n = 25)* Placebo group (n = 24) 2 sessions in the first 2 weeks and 1 session per week in the next 2 weeks	Neural techniques consisted of mobilization in craniocervical flexion, lateral cervical sliding, and opening the mouth in craniocervical flexion. The protocol lasted 15 min (5 min per technique).	Superficial massage was performed for 15 min using ultrasound gel to minimize skin stimulation.	VAS	Preintervention Post-intervention 15 days post-intervention 30 days post-intervention
Bialosky et al. [91]	n = 40 Female (n = 40) Age (years): 46.90 (10.25)	Carpal tunnel syndrome	Neural technique (n = 20) Placebo group (n = 20) 2 sessions per week for 3 weeks	A neurodynamic technique for the median nerve was applied. 5 sets of 10 cycles for the first 3 sessions and 7 sets of 10 cycles for sessions 4 through 6.	Sham group received a sham technique that minimized anatomical stress across the median nerve.	VAS	Preintervention first session Post-intervention first session Preintervention sixth session Post-intervention sixth session
Wolny and Linek [92]	n = 150 Female (n = 135) Age (years): 53.2 (SD Not available)	Carpal tunnel syndrome	Neural technique (n = 78) Placebo group (n = 72) 2 sessions per week for 10 weeks	Neurodynamic technique for the median nerve was performed in standard protocol consisting of 3 series of 60 repetitions of glide and tension neurodynamic techniques separated by inter-series intervals of 15 s.	Sham group received a sham technique that minimized anatomical stress across the median nerve.	NPRS	Preintervention Post-intervention
Fernández-Carnero et al. [93]	n = 54 Female (n = 41) Age (years): 20.91 (2.64)	Neck pain (Chronic)	Neural technique (n = 27) Placebo group (n = 27) 1 session	Neurodynamic technique for the median nerve was applied. Treatment duration was seven min (frequency of 0.5 Hz) for two min and repeated three times with 30 s of rest time between each mobilization.	Sham group received a sham technique that minimized anatomical stress across the median nerve.	VAS	Preintervention Post-intervention

Number of participants (n); Standard Deviation (SD); Numerical Rating Scale (NRS); Numerical Pain Rating Scale (NPRS); Visual Analogue Scale (VAS).

#### 3.2.4. Outcomes

NRS (16.6%) [46,48,50,53,60], NPRS (30%) [38,44,45,47,55,56,57,58,67] and VAS (53.4%) [39,40,41,42,43,49,51,52,54,59,61,62,63,64,65,66] were used in the trials for the assessment of pain. Preintervention and post-intervention measures were carried out in all studies, of which the study designs of Senna et al. [51] and Chaibi et al. [60] had the longest follow-ups, 10 months and 12 months post-intervention, respectively. All trials provided the average and standard deviation needed for the quantitative analysis, except for Kawchuck et al. [50] who did not provide any data, and Sillevis et al. [42] who did not provide the standard deviation.

#### 3.2.5. Risk of Bias’ Blinding Assessment

A summary of the results of the RoB analysis is displayed in Table 2. For the adequacy of blinding, the overall results were assessed. The trials published by Atkinson et al. [52], Pikula et al. [40], and Lehtola et al. [64] were classified as high risk in the key domain participant blinding since these studies did not simulate the intervention group procedure. In these designs, the subjects could be blinded on whether they underwent a manipulation treatment but were not blinded in the procedure of the manipulative technique. Several trials (21%) presented high risk in the detection bias domain [39,40,46,53,54], being cataloged as “Not adequate” for not meeting the required criteria. Finally, 19 trials (79%) accomplished the requirements to be considered as adequately blinded for the study group [38,41,43,44,45,47,48,49,51,52,55,56,57,58,59,62,63,66,67]. 

#### 3.2.6. Quantitative Analysis

Overall meta-analysis showed statistical significance in favor of manipulation interventions, nevertheless moderate heterogeneity was presented (SMD 0.42 (95%CI [0.23, 0.61]) *p* < 0.0001; heterogeneity [Q = 92.03; *p* < 0.0001]/[I^2^ = 65.2%]).

When the results of RoB were introduced in the subgroup analysis, statistical significance was found (Q = 4.38; *p* = 0.036), furthermore within the group cataloged as “Not adequately” blinded, the results did not find statistical significance between the intervention group and placebo control group. On the other hand, the adequately blinded group did present differences in favor of manipulation (Figure 2).

In the last subgroup analysis, analyzing different types of sham groups, the results showed no statistical significance differences between groups (Q = 1.88; *p* = 0.599). Nevertheless, it is worth mentioning, that in the forest plot analysis the only group that presented statistical differences was the simulated manipulation one, without the movement of the joints. But no differences were found in either the group that simulated applying movement, the device, or the interventions which seemed to be therapeutic (Figure 3).

### 3.3. Trials Using Soft Tissue Techniques

#### 3.3.1. Participants

In the included trials, at least one intervention group applied soft tissue techniques as a unique procedure. A total of 740 subjects were included, 515 of which were females with an overall average age of 37.49 years old. Concerning the pathologies presented by the subjects: six trials (43%) performed the intervention in subjects with neck pain [71,72,73,74,75,76], headache was prevalent in three trials (21.4%) [61,77,78], two trials (14.3%) included subjects suffering from low back pain [79,80], and chronic fatigue syndrome (7.1%) [81], lateral epicondylitis (7.1%) [82], and knee osteoarthritis (7.1%) [83] were studied in one trial, respectively.

#### 3.3.2. Intervention Groups

The soft tissue interventions applied by the trials were divided in two main techniques: massage therapy (50%) [72,74,77,78,79,80,81] and pressure release techniques on active myofascial trigger points (50%) [61,71,73,75,76,82,83]. The most common design performed only one session of intervention (50%) [71,72,73,74,75,76,83], but there were also studies which performed different number of sessions for two (14.3%) [79,82], three (21.4%) [77,80,82], four (14.3%) [61,78] and five weeks (7.1%) [81]. Inside the pressure release treatments, the most used technique was suboccipital inhibition applied in subjects suffering from neck pain and tension-type headache, however, it was performed from three min to 10 min depending on the trial. There was no consensus among the authors on the design of massage interventions.

#### 3.3.3. Placebo

As in the manipulation techniques studies, the authors followed mainly the same two strategies. In the simulation procedure, the authors who tried to simulate the technique mainly chose the gently positioning of the hands on the massage area in the absence of movement or pressure (28.6%) [73,79,82,83]; the other simulation (7.1%) consisted of faking the same treatment, applying superficial massage on the structures trying to minimize, to the biggest extent, contact with the skin [78]. Detuned devices were used as an alternative sham group, the authors applied detuned ultrasound (21.5%) [72,76,77], detuned microwave diathermy (7.1%) [74], algometer (7.1%) [75], and detuned transcutaneous electric nervous stimulation (7.1%) [81]. Three studies neither simulated procedures nor used detuned devices: Antolinos-Campillo et al. [71] performed flexion and extension of the hip for four min as a sham group against suboccipital muscle inhibition in subjects with cervical whiplash; Celenay et al. [80] performed unspecific strokes and effleurage in the lower back area, and Espí-López et al. [61] performed the artery test as placebo control for tension-type headache.

#### 3.3.4. Outcomes

For the assessment of pain in soft tissue technique trials, VAS (71.5%) [61,71,74,76,77,78,79,80,81,83], NRS (21.4%) [72,75,82] and NPRS (7.1%) [73] were used. Preintervention and post-intervention measures were assessed by all the studies, however, some authors took additional measures, the study of Arguisuelas et al. [79] conducted the longest follow-up, lasting 10 weeks. Nevertheless, three trials (21.4%) were excluded from the quantitative analysis since the authors did not present the essential data [72,73,81].

#### 3.3.5. Risk of Bias Blinding Assessment

A summary of the results of RoB is exposed in Table 2. The minimum requirements for adequately blinding the subjects were achieved by five trials (35.7%) [73,78,79,80,82]. On the other hand, nine trials were cataloged as “Not adequate” for the final analysis since six studies were cataloged as high risk in the participant blinding domain (42.8%) [71,72,74,76,77,81], given the subjects were not blinded for the soft tissue techniques procedure. High risk in allocation concealment was presented by Tanaka et al. [83], and finally, Espí-López et al. [61], was categorized as an unclear risk in more than one key domain.

#### 3.3.6. Quantitative Analysis

When the meta-analysis was performed including all the studies, medium effect size, statistical significance differences and low heterogeneity were found (SMD 0.40 (95%CI [0.19, 0.61]); *p* = 0.0017; heterogeneity [Q = 12.60; *p* = 0.32] I [I^2^ = 12.7%]).

No statistical difference was found between the adequate blinding studies and the not adequate blinding trials (Q = 0.11; *p* = 0.74). However, the trials cataloged as not adequately blinded showed statistical differences between techniques and placebo control groups, and the adequate blinding studies did not show statistical differences in favor of techniques (Figure 4).

No statistical difference was found between types of sham control groups (Q = 1.19; *p* = 0.55). Moreover, the studies that performed placebos with possible therapeutic effects did not show differences between soft tissue techniques and sham interventions. When laying hands were applied or detuned devices were used, statistical differences in pain were found between “active” and sham groups. (Figure 5).

### 3.4. Trials Using Mobilization Techniques

#### 3.4.1. Participants

A total of 544 subjects were included in manual therapy studies in which, at least, one intervention group applied isolated mobilization techniques. 327 subjects of the total subjects were females, and the average age of all the participants was 41.29 years old. Regarding the prevalence of pathologies in the subjects: two studies (25%) evaluated the neck pain [43,84], two (25%) experimented in patients suffering of low back pain [85,86], and ankle pain (12.5%) [87], cervico-craniofacial pain (12.5%) [88], thoracic pain (12.5%) [89] and shoulder pain (12.5%) [90] were studied each in one trial.

#### 3.4.2. Intervention Groups

The most common intervention (50%) consisted of one treatment session [43,84,89,90], however, La Touche et al. [88] performed six sessions in three weeks and Kogure et al. [85] carried out six sessions in six months. Regarding the mobilization emplacements, the authors mainly (75%) conducted the interventions on the vertebral spine [43,84,85,86,88,89]. Furthermore, the doses, in terms of application time and the number of repetitions, were diverse.

#### 3.4.3. Placebo

The authors for the design of sham mobilizations mostly reproduced the intervention protocol but used light force in the application of the technique with the objective of not producing a therapeutic effect on the subjects (50%) [85,87,89,90] or mimicking the grip and the procedure of mobilization without performing any force (25%) [86,88]. Both sham procedures reproduced the intervention protocol. On the contrary, Snodgrass et al. [84] did not reproduce the mobilization protocol and chose a detuned ultrasound as the sham group, and Valera-Calero et al. [43] used as the sham group, the sham manipulation group and not the sham mobilization intervention.

#### 3.4.4. Outcomes

VAS (75%) [43,84,85,87,88,90], NRS (12.5%) [89], and NPRS (12.5%) [86] were used for assessment of pain. All studies performed the evaluation before and after the treatment. Moreover, Silva et al. [87] conducted the longest follow-up, two weeks post-treatment. All trials shared the required data for the statistical analysis.

#### 3.4.5. Risk of Bias’ Blinding Assessment

A summary of the RoB results is available in Table 2. Only two studies (25%) succeeded in terms of adequately blinding [88,89]. Snodgrass et al. [84] was cataloged as “Not adequate” due to the impossibility of participant blinding since the study did not reproduce the mobilization procedure, the same as Valera-Calero et al. [43]. Also, three studies (37.5%) were cataloged as high risk in the detection bias domain [85,86,87]. Eventually, Surenkok et al. [90] also presented unclear risks in three of the four key domains.

#### 3.4.6. Quantitative Analysis

In the overall analysis, the meta-analysis did not show statistically significant difference between mobilization techniques and sham control groups (SMD 0.49 (95%CI [–0.40, 1.39]) *p* = 0.24; heterogeneity [Q = 74.62; *p* < 0.0001]/[I^2^ = 89.3%]). Neither the subgroup analysis in function of adequacy (Q = 0.81; *p* = 0.471), nor the type-of-sham subgroup analysis (Q = 0.77; *p* = 0.38) showed differences between mobilization or sham interventions (Figure 6).

### 3.5. Trials Using Neurodynamic Techniques

#### 3.5.1. Participants

From the four identified trials, a total of 341 subjects were recruited, 294 of which were females. The average age of all participants was 40.1 years old. Regarding the pathologies: two trials (50%) proved the effectiveness of neural gliding in carpal tunnel syndrome [91,92], one trial (25%) studied subjects who presented neck pain [93], and the last one (25%), included subjects who suffered from tension-type headache [78].

#### 3.5.2. Intervention Groups

The most common neurodynamic technique applied was neural gliding on the median nerve (75%) [91,92,93], however, the authors applied different dosifications according to the time of application and number of sets. Fernández-Carnero et al. [93] conducted only one session, while the other studies carried out more than one intervention session, with the Wolny et al. [92] trial being the longest, with 20 sessions in 10 weeks. Only Ferragut-Garcías et al. [78] performed neural gliding techniques in a different localization, specifically, on the cranio-cervical area.

#### 3.5.3. Placebo

All trials (75%), except for Ferragut-Garcías et al. [78], mimicked the intervention group in the same way, trying not to apply tension to the median nerve but reproducing to the maximum extent the neural mobilization technique. Ferragut-Garcías et al. [78] instead reproduced the sham neurodynamic technique by simulating the soft tissue technique intervention also presented in the trial.

#### 3.5.4. Outcomes

VAS (75%) [78,91,93] and NPRS (25%) [92] were used for assessing pain. All trials performed preintervention and post-intervention measures. Although Ferragut-Garcías et al. [78] was the only study that completed a 10-week follow-up.

#### 3.5.5. Risk of Bias’ Blinding Assessment

A summary of the RoB results is available in Table 2. The trial of Ferragut-Garcías et al. [78] was the only one that was cataloged as inadequately blinded since they did not secure the blinding participant because they did not reproduce the neurodynamic technique procedure. The rest of the trials (75%) were cataloged as successful overall for adequately blinding [91,92,93].

#### 3.5.6. Quantitative Analysis

The meta-analysis of the studies included did not show statistical differences and high heterogeneity was found (SMD 0.99 (95%CI [–0.74, 2.73]) *p* = 0.16; heterogeneity [Q = 58.62; *p* < 0.0001]/[I^2^ = 94.9%]). Since only four studies were included, it was not possible to perform the subgroup analysis (Figure 7).

### 3.6. Quality of Evidence

After the assessment of the quality of evidence through GRADE, the level of recommendation for manipulation techniques and soft tissue techniques was low for the management of pain. In the case of mobilization techniques and neurodynamic techniques it was very low (Table 3).

## 4. Discussion

The results revealed that it is not clear whether manual therapy techniques are superior to sham control groups. The main reason may reside in the use of multiple designs of sham groups found in the literature. When different sham groups were divided into subgroups, less heterogeneity was found, and some conclusions could be made.

Throughout the last decades, the placebo effect has been the subject of numerous studies by the scientific community. However, in the field of manual therapy, the placebo effect has been undervalued on numerous occasions, probably as a result of the difficulty of blinding the participants, the lack of reliable interventions, and the inability of blinding the personnel (responsible for applying the interventions). In 2017, Bialosky et al. [94] questioned the effectiveness of the techniques of manual therapy and how many of those possible outcomes were due to the placebo effect. The evaluation of blinding success (for the subject or evaluator) has been a requirement of guidelines to design quality sham groups [95,96,97]. The blinding index has proved to be an interesting tool to properly assess blinding in RCTs. Furthermore, incorporating this tool in manual therapy studies [98,99] could provide factual information on the perception of the subjects who underwent a research intervention. Regarding the blinding of the personnel, it is convenient to note that although this factor could influence the outcomes of the studies [100], in this analysis it was assumed that this requirement in manual therapy trials would be impossible to achieve.

Nonetheless, it would be important to point out that sham groups are not widespread and standardized in the literature. Manipulation is the technique most researchers use to seek plausible sham groups; authors such as Vernon et al. [44,70], Chaibi et al. [101], and Michener et al. [69,102] have designed protocols for the development of valid sham groups. In this regard, the results are along the same lines as other reviews that evaluated sham groups [19,22]. Vernon et al. [24] and Puhl et al. [26] assessed the quality of sham groups in trials of cervical manipulation, and lumbar and pelvic manipulation, respectively, both systematic reviews concluded the lack of quality sham control groups. Metanalyses were also carried out assessing the different types of placebo control groups for manipulation techniques in chronic back pain [103] and chronic low back pain [104], nevertheless, contradictory results were found. In addition, Hancock et al. [25] questioned 25 experts on which sham procedure could be more feasible to control manipulation RCTs, but amongst the participants, there was an extremely low level of agreement.

When comparing the studies according to the adequacy of blinding evaluated by the RoB, active manipulation techniques showed significantly bigger pain reduction when compared to sham groups in trials with adequate blinding. Nevertheless, in the subgroup analysis involving several types of sham, the only studies which simulated the technique without applying movement-active treatment showed significantly better pain reduction than the sham group. The studies which chose a detuned device or simulated the high-speed movement did not show statistical differences between active and sham groups. This contradiction could be explained by the expectations generated by the different control groups; in other words, it is possible that the positive results of the manipulation in pain management may be caused by the positive expectations of the subjects.

Contrary to the last analysis, and also beholding RoB analysis, in the soft tissue techniques, the studies that were adequately blinded did not show statistically significant differences between active and sham groups, while studies that were inadequately blinded showed a statistical difference in favor of soft tissue techniques. However, when the analysis was carried out in the function of the types of sham control groups, the control groups which had potential therapeutic effects did not show statistical differences, but the detuned device group and the simulation group did. Therefore, soft tissue techniques presented positive effects on pain management. Recent reviews share similar results in terms of the effectiveness of myofascial release [105] and ischemic compression [106] techniques in the management of myofascial pain syndrome.

As for the mobilization techniques and the neurodynamic techniques, the limited number of studies constrained our analysis. No significant differences were found between groups for any of the analyses performed, nor were we able to perform the analysis in the function of the types of placebos. Further studies comparing neurodynamic techniques and mobilization are needed. Recent investigations have found that neural mobilization is useful to manage neck pain [107] and back pain [107,108], however higher quality RCT could be clarifying.

The aim of placebo control groups is to generate the same expectations in the subjects as the expectations generated by the “active” intervention groups [18].

Also, it is worthwhile to highlight that all these results support the exclusion from this work of studies that used detuned devices as placebo groups for not implementing a reliable sham control group for manual therapy techniques. Nevertheless, when analyzing the results from different sham procedures, detuned devices obtained similar results as the other sham procedures. For future research, it could be interesting to evaluate the expectations of the different sham control groups and how that affects treatment effect estimates.

### Study Limitations

The study presents several limitations. Firstly, the lack of studies found that compared manual therapy with sham control groups. It may be due to the difficulty and the low consensus when authors have to develop quality placebos. Furthermore, another limitation is the length of the interventions, since the number and the frequency of the sessions were not taken into consideration, causing dropouts was not discussed, as in previous studies [109]. Nor were the characteristics of the participants such as sex, age, or pain etiology compared. Lastly, only studies written in English were included. The effect of blinding was evaluated following similar guidelines as previous studies [31]; however, blinding could be analyzed in different ways and thus future research should challenge this evaluation and perhaps perform sensitivity analyses for different types of blinding when using sham groups.

## 5. Conclusions

In summary, the literature presents a lack of a unified placebo control group design for each technique and an absence of expectations assessed. These two issues might account for the unclear results obtained in the analysis. The manipulation techniques were demonstrated to be more effective in pain reduction than placebo control groups in the overall analysis. However, manipulation techniques did not show superior effectiveness when compared with all types of placebos, raising doubts about their therapeutic effect which could be resolved in future studies with the evaluation of participants’ expectations in the different sham groups, therefore the efficacy could not be concluded in this study. In the case of soft tissue techniques, the results are stronger in favor of these techniques when compared to placebo control groups, being the soft tissue techniques more effective than placebo control groups for the management of pain. Even so, the authors suggest the same recommendations, the evaluation of participants’ expectations. Finally, regarding the mobilization techniques and neural gliding techniques, not enough studies were found in order to make conclusions.

## Figures and Tables

**Figure 1 ijerph-19-14021-f001:**
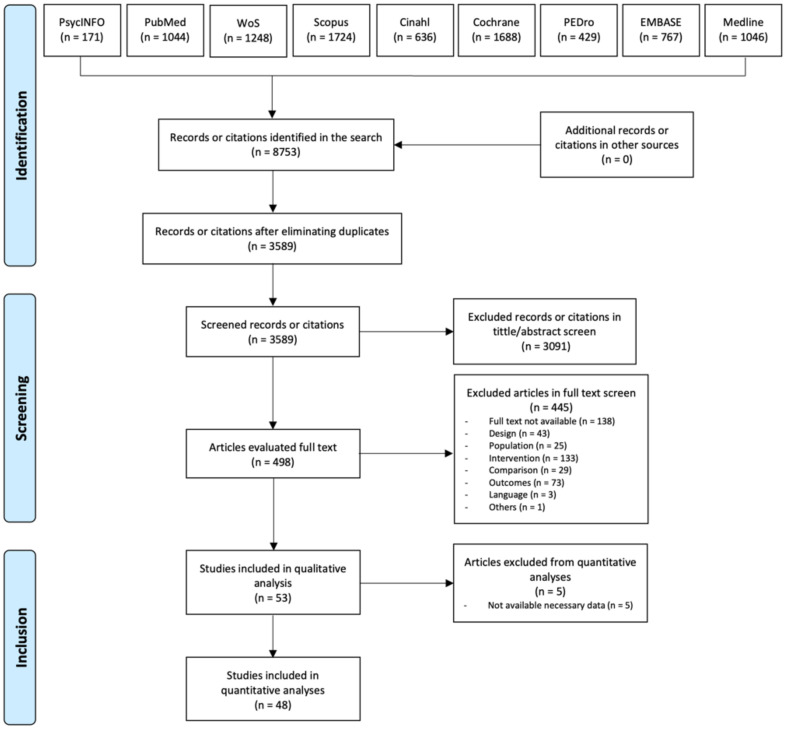
Preferred Reporting Items for Systematic Reviews and Meta-Analyses (PRISMA) flow diagram.

**Figure 2 ijerph-19-14021-f002:**
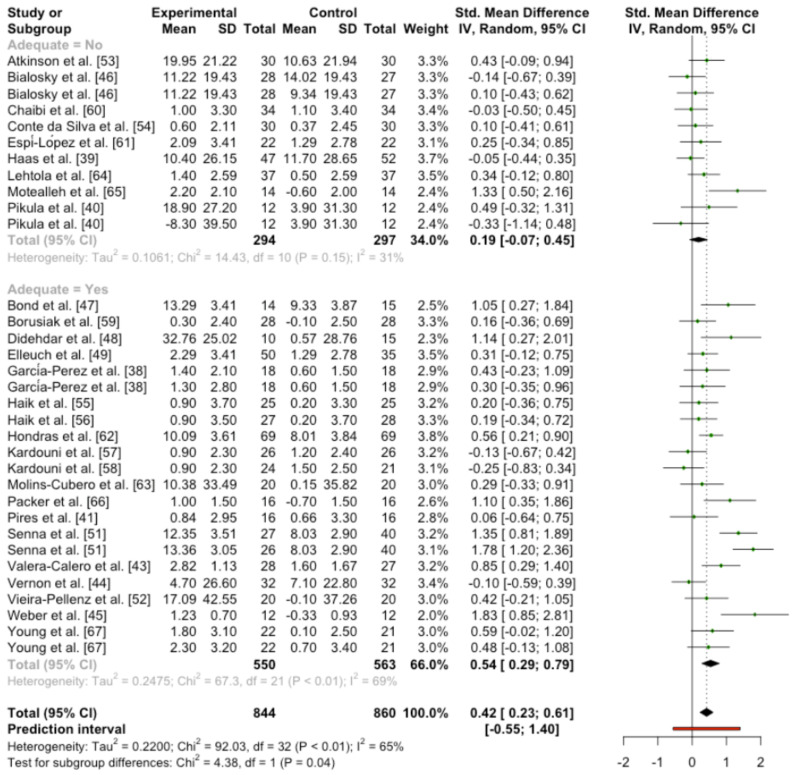
Risk of Bias Subgroup Analysis of manipulation trials. Positive values = In favor of intervention/Negative values = In favor of sham intervention. Standard Deviation (SD); Standard (std); Inverse Variance (IV); Confidence Interval (CI).

**Figure 3 ijerph-19-14021-f003:**
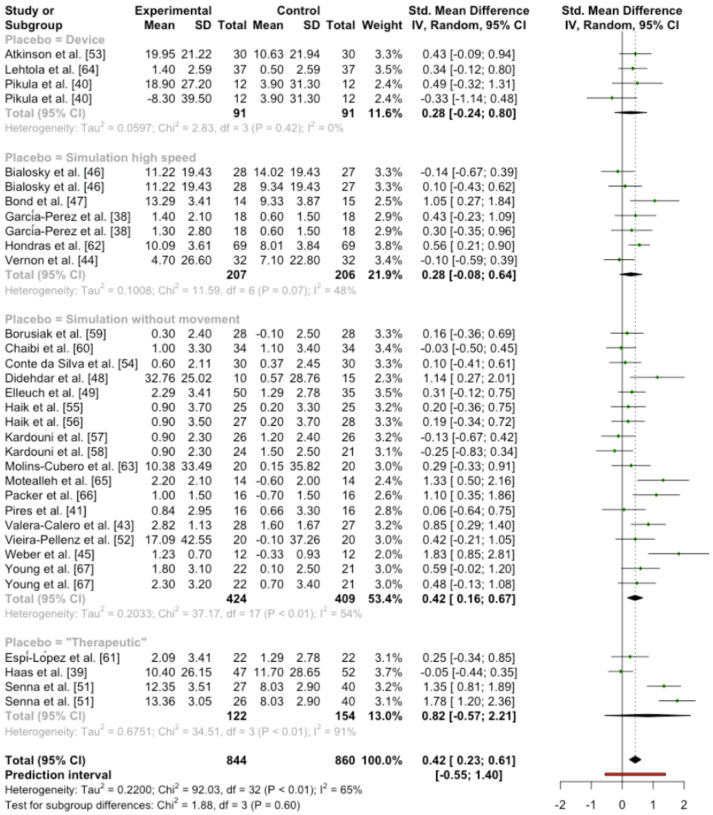
Types of Placebo Subgroup Analysis of manipulation trials. Positive values = In favor of intervention/Negative values = In favor of sham intervention. Standard Deviation (SD); Standard (std); Inverse Variance (IV); Confidence Interval (CI).

**Figure 4 ijerph-19-14021-f004:**
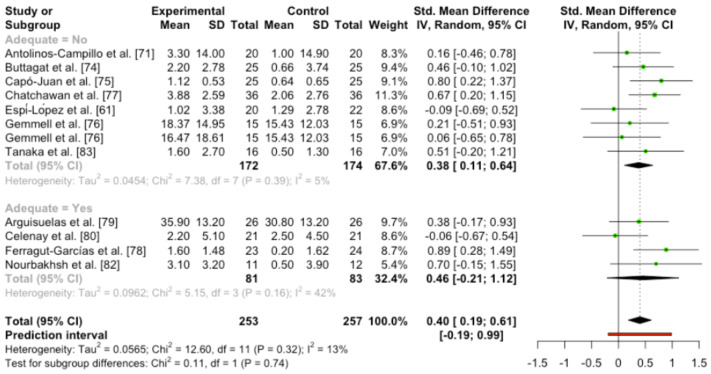
Risk of Bias Subgroup Analysis of soft tissue techniques trials. Positive values = In favor of intervention/Negative values = In favor of sham intervention. Standard Deviation (SD); Standard (std); Inverse Variance (IV); Confidence Interval (CI).

**Figure 5 ijerph-19-14021-f005:**
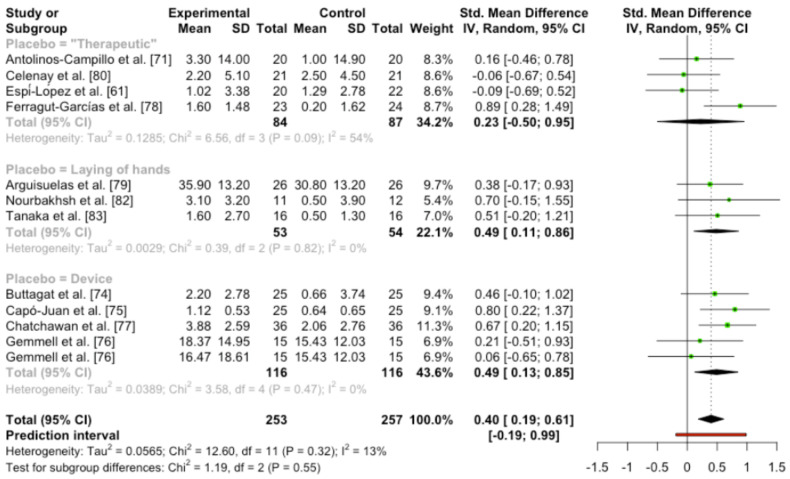
Types of Placebo Subgroup Analysis of soft tissue techniques trials. Positive values = In favor of intervention/Negative values = In favor of sham intervention. Standard Deviation (SD); Standard (std); Inverse Variance (IV); Confidence Interval (CI).

**Figure 6 ijerph-19-14021-f006:**
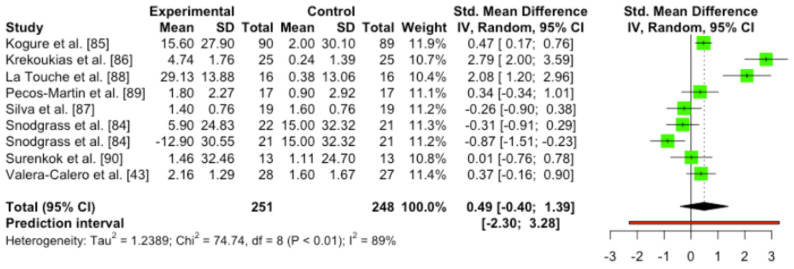
Meta-analysis of mobilization trials. Positive values = In favor of intervention/Negative values = In favor of sham intervention. Standard Deviation (SD); Standard (std); Inverse Variance (IV); Confidence Interval (CI).

**Figure 7 ijerph-19-14021-f007:**
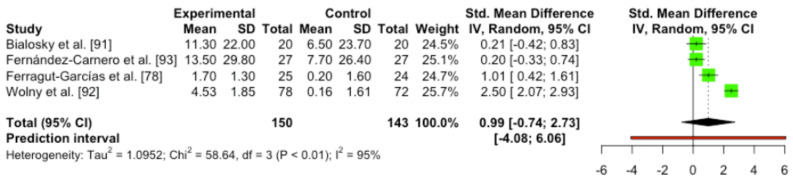
Meta-analysis of neural gliding trials. Positive values = In favor of intervention/Negative values = In favor of sham intervention. Standard Deviation (SD); Standard (std); Inverse Variance (IV); Confidence Interval (CI).

**Table 2 ijerph-19-14021-t002:** Cochrane Risk of Bias tool for randomized trials. Risk of bias assessment (n = 53 trials).

**Manipulation Tecniques**											
**Author **	**Selection Bias**		**Performance Bias**			**Detection Bias**	**Attrition Bias**	**Reporting Bias**	**Other Bias**	**OVERALL**	**Adequated**
	**Random** **Sequence Generation**	**Allocation Concealment**	**Participant** **Blinding**	**Research** **Personnel Blinding**	**Therapist Blinding**						
Atkinson et al. [53]	x	?	x	?	x	x	x	✓	x	x	No
Bialosky et al. [46]	✓	✓	✓	?	x	x	✓	✓	✓	x	No
Bond et al. [47]	✓	✓	✓	?	x	✓	✓	✓	✓	✓	Yes
Borusiak et al. [59]	✓	✓	✓	✓	x	✓	x	✓	✓	✓	Yes
Chaibi et al. [60]	✓	?	✓	?	x	?	x	✓	✓	?	No
Conte-da-Silva et al. [54]	✓	✓	✓	✓	x	x	✓	✓	✓	x	No
Didehdar et al. [48]	✓	?	✓	?	x	✓	x	✓	x	✓	Yes
Elleuch and Ghroubi [49]	✓	?	✓	?	x	✓	x	✓	x	✓	Yes
Espí-López et al. [61]	✓	?	?	?	x	?	✓	✓	✓	x	No
García-Perez-Juana et al. [38]	✓	✓	✓	?	x	✓	✓	✓	✓	✓	Yes
Haas et al. [39]	✓	✓	✓	✓	✓	x	✓	✓	✓	x	No
Haik et al. [55]	✓	?	✓	?	x	✓	✓	✓	✓	✓	Yes
Haik et al. [56]	✓	?	✓	?	x	✓	✓	✓	✓	✓	Yes
Hondras et al. [62]	✓	✓	✓	?	x	✓	✓	✓	✓	✓	Yes
Kardouni et al. [57]	✓	✓	✓	?	x	✓	✓	✓	✓	✓	Yes
Kardouni et al. [58]	✓	✓	✓	?	x	✓	✓	✓	✓	✓	Yes
Kawchuk et al. [50]	?	?	✓	?	x	✓	✓	x	✓	?	No
Lehtola et al. [64]	✓	?	x	?	x	✓	✓	✓	✓	x	No
Molins-Cubero et al. [63]	✓	✓	✓	?	x	✓	✓	✓	✓	✓	Yes
Motealleh et al. [65]	✓	?	?	?	x	?	✓	✓	x	x	No
Packer et al. [66]	?	✓	✓	✓	x	✓	✓	✓	✓	✓	Yes
Pikula [40]	✓	?	x	?	x	x	✓	✓	✓	x	No
Pires et al. [41]	?	✓	✓	?	x	✓	✓	✓	✓	✓	Yes
Senna and MacHaly [51]	✓	✓	✓	?	x	✓	x	✓	x	✓	Yes
Sillevis et al. [42]	✓	✓	?	?	x	?	✓	✓	✓	?	No
Valera-Calero et al. [43]	✓	✓	✓	?	x	✓	✓	✓	✓	✓	Yes
Vernon et al. [44]	✓	✓	✓	?	x	?	✓	✓	✓	✓	Yes
Vieira-Pellenz et al. [52]	✓	?	✓	?	x	✓	✓	✓	✓	✓	Yes
Weber et al. [45]	✓	?	✓	?	x	✓	✓	✓	✓	✓	Yes
Young et al. [67]	✓	✓	✓	?	x	✓	✓	✓	✓	✓	Yes
**Soft Tissue Techniques**											
**Author**	**Selection Bias**		**Performance Bias**			**Detection Bias**	**Attrition Bias**	**Reporting Bias**	**Other Bias**	**Overall**	**Adequated**
	**Random** **Sequence Generation**	**Allocation Concealment**	**Participant Blinding**	**Research** **Personnel Blinding**	**Therapist** **Blinding**						
Antolinos-Campillo et al. [71]	✓	?	x	✓	x	✓	✓	✓	✓	?	No
Arguisuelas et al. [79]	✓	?	✓	?	x	✓	✓	✓	✓	✓	Yes
Blikstad and Gemmell [72]	✓	✓	x	?	x	✓	✓	?	x	x	No
Briem et al. [73]	✓	✓	✓	?	x	✓	✓	x	✓	✓	Yes
Buttagat et al. [74]	✓	✓	x	✓	x	✓	✓	✓	✓	x	No
Capó-Juan et al. [75]	✓	?	?	✓	x	?	✓	✓	✓	x	No
Celenay et al. [80]	✓	?	✓	?	x	✓	✓	✓	✓	?	Yes
Chatchawan et al. [77]	✓	✓	x	✓	x	✓	✓	x	✓	x	No
Espí-López et al. [61]	✓	?	?	?	x	?	✓	✓	✓	x	No
Ferragut-Garcías et al. [78]	✓	✓	✓	?	x	✓	✓	✓	✓	✓	Yes
Field et al. [81]	✓	?	x	✓	x	✓	?	✓	✓	x	No
Gemmell et al. [76]	✓	✓	x	?	x	✓	✓	✓	✓	x	No
Nourbakhsh and Fearon [82]	✓	?	✓	?	x	✓	?	✓	✓	✓	Yes
Tanaka et al. [83]	✓	x	?	?	x	✓	✓	✓	✓	x	No
**Mobilization Techniques**											
**Author**	**Selection Bias**		**Performance Bias**			**Detection Bias**	**Attrition Bias**	**Reporting Bias**	**Other Bias**	**Overall**	**Adequated**
	**Random** **Sequence Generation**	**Allocation Concealment**	**Participant Blinding**	**Research** **Personnel Blinding**	**Therapist Blinding**						
Kogure et al. [85]	✓	?	✓	x	x	x	✓	✓	✓	x	No
Krekoukias et al. [86]	✓	?	✓	?	x	x	✓	✓	✓	x	No
La Touche et al. [88]	✓	?	✓	?	x	✓	✓	✓	✓	✓	Yes
Pecos-Martin et al. [89]	?	✓	✓	?	x	✓	?	✓	✓	✓	Yes
Silva et al. [87]	x	?	✓	✓	x	x	✓	✓	✓	x	No
Snodgrass et al. [84]	✓	✓	x	?	x	✓	✓	✓	✓	x	No
Surenkok et al. [90]	?	?	?	?	x	✓	✓	✓	✓	x	No
Valera-Calero et al. [43]	✓	✓	x	?	x	✓	✓	✓	✓	x	No
**Neural Techniques**											
**Author**	**Selection Bias**		**Performance Bias**			**Detection Bias**	**Attrition Bias**	**Reporting Bias**	**Other Bias**	**Overall**	**Adequated**
	**Random** **Sequence Generation**	**Allocation Concealment**	**Participant Blinding**	**Research** **Personnel Blinding**	**Therapist Blinding**						
Bialosky et al. [91]	✓	✓	✓	?	x	✓	✓	✓	✓	✓	Yes
Fernández-Carnero et al. [93]	✓	✓	✓	?	x	✓	✓	✓	✓	✓	Yes
Ferragut-Garcías et al. [78]	✓	✓	x	?	x	✓	✓	✓	✓	✓	No
Wolny and Linek [92]	✓	?	✓	?	x	✓	x	✓	✓	✓	Yes

Low risk (✓); Unclear risk (?); High risk (x).

**Table 3 ijerph-19-14021-t003:** GRADE assessment.

GRADE Assessment							
Outcomes	Study Design	Risk of Bias −1 Serious −2 Very Serious	Inconsistency −1 Serious −2 Very Serious	Indirectness −1 Serious −2 Very Serious	Imprecision −1 Serious −2 Very Serious	Large Effect +1 Large +1 Very Large	Quality of Evidence
Manipulation techniques	RCT	−1	−1	0	0	0	Low
Soft tissue techniques	RCT	−2	0	0	0	0	Low
Mobilization techniques	RCT	−2	−2	0	−1	0	Very low
Neurodynamic techniques	RCT	−1	−2	0	−1	1	Very low

Randomized Clinical Trial (RCT).

## Data Availability

Not applicable.

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
