# Peer review of "Manual Therapy Effect in Placebo-Controlled Trials: A Systematic Review and Meta-Analysis"

_ijerph, 2022, doi:10.3390/ijerph192114021_

Round 1

Reviewer 1 Report

This paper systematically reviewed and meta-analysed data available on the manual therapy effect in placebo-controlled trials. This is an important area of research. However, there are some concerns with the paper in its current form. Please see below my comments.

Abstract: Please add the analysis results to the result section

Why did you include all of the diseases? This makes it difficult to comment.

What was the difference between your study from other reviews? “Sham treatment effects in manual therapy trials on back pain patients: a systematic review and pairwise meta-analysis. BMJ Open. 2021 May 4;11(5):e045106. doi: 10.1136/bmjopen-2020-045106” Also, please add this study to the discussion section.

“Ruddock JK, Sallis H, Ness A, et al. Spinal manipulation vs sham manipulation for nonspecific low back pain: a systematic review and meta-analysis. J Chiropr Med 2016;15:165–83.”

“Rubinstein SM, de Zoete A, van Middelkoop M, et al. Benefits and harms of spinal manipulative therapy for the treatment of chronic low back pain: systematic review and meta-analysis of randomised controlled trials. BMJ 2019;364:l689”

Further, there is no discussion about adverse events or Drop-outs.

Finally, there is no attempt to position this paper in the context of what exists in the literature and what is needed next in the field, which is important given the literature base available for Manuel therapy in this population.

Author Response

International Journal of Environmental Research and Public Health:

We appreciate the opportunity to revise our manuscript ijerph-1963952, updated and entitled “Manual Therapy Effect in Placebo-Controlled Trials: A Systematic Review and Meta-Analysis" to International Journal of Environmental Research and Public Health. We trust that you will find the current version informative to your readership and acceptable for publication.

Thanks for your review commentaries in order to improve the quality of the manuscript. A deep and substantial modification has been carried out according to your suggestions.

Modifications to the manuscript text are denoted by line numbers and yellow highlighted.

Please find our responses to each reviewers comment below.

                                                                               Sincerely,          

                                                                               Josué Fernández-Carnero

Responses to Reviewers' comments:

Reviewer #1:

Thanks for your comments to improve the quality of the manuscript. Deep and substantial modifications have been carried out according to your suggestions.

Please see below our answers.

Abstract: Please add the analysis results to the result section

Response: Standard Mean Difference (SMD) and Confident Intervals (CI) have been added in lines 34-38, illustrating the most important highlights. To do so, for the manipulation results, we added the SMD and CI of all the subgroups “type of placebo” comparisons, with the data presented there were no statistical differences in 3 of 4 comparisons since the 0 was included in the Confident Interval. For the soft tissue techniques we included de SMD and CI of the metanalysis. Finally, due to the lack of studies found in the groups of mobilization techniques and neural gliding, we considered better not to include the data that could lead to misunderstanding. Thank you again for your suggestion.

Why did you include all of the diseases? This makes it difficult to comment.

Response: Thank you for your question, we fully understand your observation. We wanted to evaluate whether the design of placebo control groups could produce different interpretations of the efficacy of manual therapy techniques, independently of the subjacent pathology. We believe that a wide-open search could lead to a better understanding of the placebo control groups and how it could modify the interpretation of the manual therapy effectiveness in terms of pain. We were then unaware of whether the placebo control groups were “well” designed for some pathology or part of the body, and that could help to future researchers of other areas of investigation.  We hope you this response meets your expectations.

What was the difference between your study from other reviews? “Sham treatment effects in manual therapy trials on back pain patients: a systematic review and pairwise meta-analysis. BMJ Open. 2021 May 4;11(5):e045106. doi: 10.1136/bmjopen-2020-045106” Also, please add this study to the discussion section.

“Ruddock JK, Sallis H, Ness A, et al. Spinal manipulation vs sham manipulation for nonspecific low back pain: a systematic review and meta-analysis. J Chiropr Med 2016;15:165–83.”

“Rubinstein SM, de Zoete A, van Middelkoop M, et al. Benefits and harms of spinal manipulative therapy for the treatment of chronic low back pain: systematic review and meta-analysis of randomised controlled trials. BMJ 2019;364:l689”

Response: Thank you for the question. The three studies are very well designed, however deep differences in the objectives exist between the three studies and when compared with our study. The main difference with these articles is that our review did not only focus on back pain (Lavazza et al.) and low back pain (Rubinstein et al., Ruddock et al.), we included such pathologies, but we also included every pathology that could potentially be beneficiated of manual therapy. Lavazza et al. only compared with sham procedures but included techniques as reflexology or holistic treatments that we did not include because of the difficulty of homogenization for the qualitative and specially the quantitative analysis.

In Rubinstein et al., the control group was not always a placebo control group, since they added as comparators: recommended interventions, non-recommended interventions, and combination therapy. Their aim was to compare benefits against harms of manipulation techniques in patients with chronic low back pain, not to see the differences between different types of placebo control groups and how they could affect the effectivity of the technique.

Finally, Ruddock et al. performed a very similar analysis, nevertheless they only included manipulation techniques and participants with low back pain. Also, they tried to include only credible sham procedures, however they did not analyze if the different types of placebo control groups could influence in the interpretation of the effectiveness of manipulation.

These articles have been included in the discussion, thank you for the suggestion.

Further, there is no discussion about adverse events or Drop-outs.

Response: Thank you for the comment.

We did not include adverse events because the aim of the study was to focus more on the placebo control group than in the manual therapy technique itself. We assumed that the techniques were effective to the manage of pain and we wanted to see if they were effective because of the technique itself or because the placebo effect.

On the other hand, we are aware that one of the main limitations of the article was not taking into consideration the number and the frequency of the sessions, this aspect was reflected and included in such a part of the manuscript. We chose to perform bigger analyses in order to get more power in the analyses, moreover we did not focus on long time effects, neither in the number of sessions, that could cause drop-outs. Included in lines 531-533.

We sincerely hope that you understand.

Finally, there is no attempt to position this paper in the context of what exists in the literature and what is needed next in the field, which is important given the literature base available for Manuel therapy in this population.

Response: Thank you for the comment. We tried to explain the relevance of this study in the discussion, but from another point of view. We humbly know that with the present design we could not conclude whether the manual therapy is effective for the management of pain. On the other hand, we wanted to share with our colleagues the importance of the design of placebo control groups and how it can influence how we interpret the effectiveness of a technique. We are aware that creating a “well designed” placebo control group is complicated, nevertheless we need to work in the same direction and develop standardized ways of design placebo control groups, as we included in the discussion, nowadays this consensus still inexistent. (Hancock, M.J.; Maher, C.G.; Latimer, J.; et al. Selecting an appropriate placebo for a trial of spinal manipulative therapy. Aust. J. Physiother. 2006, 52, 135–138. DOI: 10.1016/s0004-9514(06)70049-6)

Reviewer 2 Report

Dear Authors. I appreciate that the review you have has been done.

Since placebo treatment in manual therapy it was always difficult to perform and there are many criticism in doing that during the RCT, the review do not include the majority of the studies that compare manula therapy with physical therapies . For this reason conclusion should emphasize that the inefficiency of manual therapy can not be defined by this study. Conversely, a some more comments could be made in relation to the soft tissue studies where the results were stronger. Thanks

Author Response

Reviewer #2:

Thanks for your commentaries to improve the quality of the manuscript. A deep and substantial modification has been carried out according to your suggestions.

Please see below our answers.

Since placebo treatment in manual therapy it was always difficult to perform and there are many criticism in doing that during the RCT, the review do not include the majority of the studies that compare manula therapy with physical therapies . For this reason conclusion should emphasize that the inefficiency of manual therapy can not be defined by this study. Conversely, a some more comments could be made in relation to the soft tissue studies where the results were stronger. Thanks

Response: Thank you very much for the comments. We have added those suggestions to the paper in lines 589-595.

Reviewer 3 Report

A very interesting topic, indicating the need for research on the topic raised.

Verse 110 - The authors wrote: ‘The following databases were searched until 11 December 2019, from inception”, we are now in 2022, i.e. the authors have not included the time 2019-2022, they should complete the review by 2022.

I encourage authors to review publications up to 2022, perhaps they will find something else interesting for their manuscript.

Perhaps a review by 2022 could prove crucial in the subject matter undertaken.

Author Response

Reviewer #3:

Thanks for your commentaries in order to improve the quality of the manuscript. Please see below our answers.

A very interesting topic, indicating the need for research on the topic raised.

Verse 110 - The authors wrote: ‘The following databases were searched until 11 December 2019, from inception”, we are now in 2022, i.e. the authors have not included the time 2019-2022, they should complete the review by 2022.

I encourage authors to review publications up to 2022, perhaps they will find something else interesting for their manuscript.

Perhaps a review by 2022 could prove crucial in the subject matter undertaken.

Response: Thank you very much for the comment. We seriously considered to update the review, however we found pros and cons, making this improvement very controversial. The systematic review was registered in PROSPERO including the date of the search, so updating the review could be a red flag. Also, in the last version of the Cochrane Handbook for Systematic Reviews of Interventions (version 6.3, 2022) the authors concluded that the median time to require an update was 5.5 years, and we are not yet at that point.

However, the update might allow us to find some articles that could help us to improve the review.

We have considered better not to update the review trying to avoid falling in methodological issues, however if you find it mandatory, we will be happy to update it.

Round 2

Reviewer 1 Report

All my previous concerns and questions were addressed and answered